inorganic chemistry/materials science/energy

zeolitic imidazolate frameworks, hierarchical Porous ZIF-8, dye encapsulation, dye-sensitized solar cells, photophysical properties, emission lifetime

**Authors for correspondence:**
Hani Nasser Abdelhamid
e-mail: hany.abdelhameed@science.au.edu.eg,
chemist.hani@yahoo.com
Jiayan Cong
e-mail: jiayan@kth.se
Xiaodong Zou
e-mail: xzou@mmk.su.se

This article has been edited by the Royal Society of Chemistry, including the commissioning, peer review process and editorial aspects up to the point of acceptance.

# Towards implementing hierarchical porous zeolitic imidazolate frameworks in dye-sensitized solar cells

Hani Nasser Abdelhamid[1,2], Ahmed M. El-Zohry[3], Jiayan Cong[4], Thomas Thersleff[1], Martin Karlsson[4], Lars Kloo[4] and Xiaodong Zou[1]

[1]Department of Materials and Environmental Chemistry, Stockholm University, Stockholm 106 91, Sweden
[2]Advanced Multifunctional Materials Laboratory, Department of Chemistry, Assiut University, Assiut 71515, Egypt
[3]Department of Chemistry, Ångström Laboratories, Uppsala University, PO Box 523, 75120 Uppsala, Sweden
[4]Applied Physical Chemistry, Department of Chemistry, KTH Royal Institute of Technology, Teknikringen 30, 10044 Stockholm, Sweden

HNA, 0000-0002-3106-8302

A one-pot method for encapsulation of dye, which can be applied for dye-sensitized solar cells (DSSCs), and synthesis of hierarchical porous zeolitic imidazolate frameworks (ZIF-8), is reported. The size of the encapsulated dye tunes the mesoporosity and surface area of ZIF-8. The mesopore size, Langmuir surface area and pore volume are 15 nm, $960–1500\ m^2 \cdot g^{-1}$ and $0.36–0.61\ cm^3 \cdot g^{-1}$, respectively. After encapsulation into ZIF-8, the dyes show longer emission lifetimes (greater than 4–8-fold) as compared to the corresponding non-encapsulated dyes, due to suppression of aggregation, and torsional motions.

## 1. Introduction

Solar energy is a promising alternative energy source to replace traditional fossil fuels [1–3]. Dye-sensitized solar cells (DSSCs, or Grätzel cells) represent a third generation solar cell technology, characterized by the potential for low production costs, green technologies and high conversion efficiency [1–4]. In DSSCs, the dye chromophore has the central task of absorbing light to generate electrons that are transferred to the sensitized semiconductor substrate typically on an fs-ns time scale or alternatively are lost through dissipation via radiative relaxation or internal conversion (IC) [5–9]. The structure–property

relationship [10,11] for a sensitizing dye depends on a series of properties, such as anchoring group [12] and fluorescence lifetime [13]. The aggregation and torsional motions of the dyes decrease the efficiency of the measured cell. Encapsulating the dyes into porous materials may reduce both aggregation and torsional motions of the dye.

Metal-organic frameworks (MOFs) are self-assembled porous materials with potential for several applications [14–23], including DSSCs [24–27]. Both the Materials of Institute Lavoisier-125 (MIL-125), and zeolitic imidazolate frameworks (ZIF-8) have shown exceptional enhancement of open circuit voltage ($V_{OC}$), and the electron lifetime compared to other reported MOFs [24]. ZIF-8 showed good electrochemical properties, and improved the cell photo-current response attributed to reduction of recombination charge losses [28,29]. ZIF-8 shows great potential for DSSC development [30]. Dye encapsulated into MOFs is a promising approach to improve photophysical properties of the encapsulated dye [31–36], and exhibits a broadband white emission [37], fast response, high photostability and tunable sensitivity [38]. The encapsulation of a dye into an MOF prevents the dye molecules from aggregation-caused quenching, and increases the efficiency of energy transfer to the dye [31]. The synthesis procedures of ZIF-8 can be used for *in-situ* loading of dyes [39]. Furthermore, the dyes can also be used as a template for the synthesis of hierarchical porous ZIF-8 (micro-and mesopore structure), and tune the pore size [39].

Herein, a method for encapsulation of dyes into ZIF-8 crystals is reported for application in DSSCs. The materials were synthesized using a one-pot method, and resulted in hierarchical porous structures of ZIF-8 [39]. The materials were characterized using X-ray diffraction (XRD), transmission electron microscopy (TEM), high angle annular dark field (HAADF), electron energy loss spectroscopy (EELS), scanning electron microscopy (SEM), $N_2$ adsorption–desorption isotherms, fluorescence spectroscopy and emission lifetime measurements. The lifetime measurements reveal an increase on the dye's excited state upon encapsulation due to suppression of both aggregation and torsional motions. The materials were also tested for DSSCs.

# 2. Material and methods

L1 (5-[4-(diphenylamino)phenyl]thiophene-2-cyanoacrylic acid) is a commercially available dye, while the dyes L1Fc and L1Fc$_2$ were synthesized as described in ref. [6]. Zn(NO$_3$)$_2$·6H$_2$O, triethylamine (TEA) and 2-methylimidazole (Hmim) were purchased from Sigma (Germany).

## 2.1. Synthesis of dye@ZIF-8

Solutions of the dyes (L1, L1Fc and L1Fc$_2$, figure 1a) were prepared in methanol (1 mg ml$^{-1}$). Solution of Zn(NO$_3$)$_2$·6H$_2$O (0.84 M) and Hmim (3 M) were prepared in water.

ZIF-8 and dye@ZIF-8 were synthesized in a scintillation vial using triethylamine-assisted method with modification [39]. The dye is insoluble in water; thus, it was dissolved in methanol. Triethylamine (TEA, 0.7 mmol (0.1 ml), or 7 mmol (1 ml)) was added to a solution of Zn(NO$_3$)$_2$·6H$_2$O (0.7 mmol, 0.8 ml). A solution of the dye (L1, LIFc or L1Fc$_2$, 1 ml) was added, followed by the addition of 8 ml of the Hmim solution. The reaction volume was completed to 17 ml using deionized water. The reaction mixture was stirred for 30 min. The particles were collected using centrifugation (13 500 r.p.m., 30 min). The materials were washed using water and ethanol (2 × 25 ml). The yields of the prepared materials were 80–98% (electronic supplementary material, table S1).

Conventional microporous ZIF-8 without any dyes was synthesized in [39]. L1 adsorbed into ZIF-8 is performed via soaking 15 mg of microporous ZIF-8 into the dye solution (15 ml, 1 mg ml$^{-1}$) for 12 h before removing the dye solution and washing using water (2 × 25 ml), and ethanol (2 × 25 ml).

Preparation of dye@ZIF-8-coated TiO$_2$ electrodes, instruments used for characterization, electrochemical impedance spectroscopy, evaluation of DSSCs parameters [40] and time correlated single photon counting (TCSPC) experiments are described in the electronic supplementary material.

# 3. Results and discussion

## 3.1. Characterization of dye@ZIF-8

The dyes used in the solar cells (L1, L1Fc, L1Fc$_2$, figure 1a) were encapsulated into ZIF-8 using a one-pot method [39]. The presence of methanol in the reaction mixture increases the growth rate of the ZIF-8

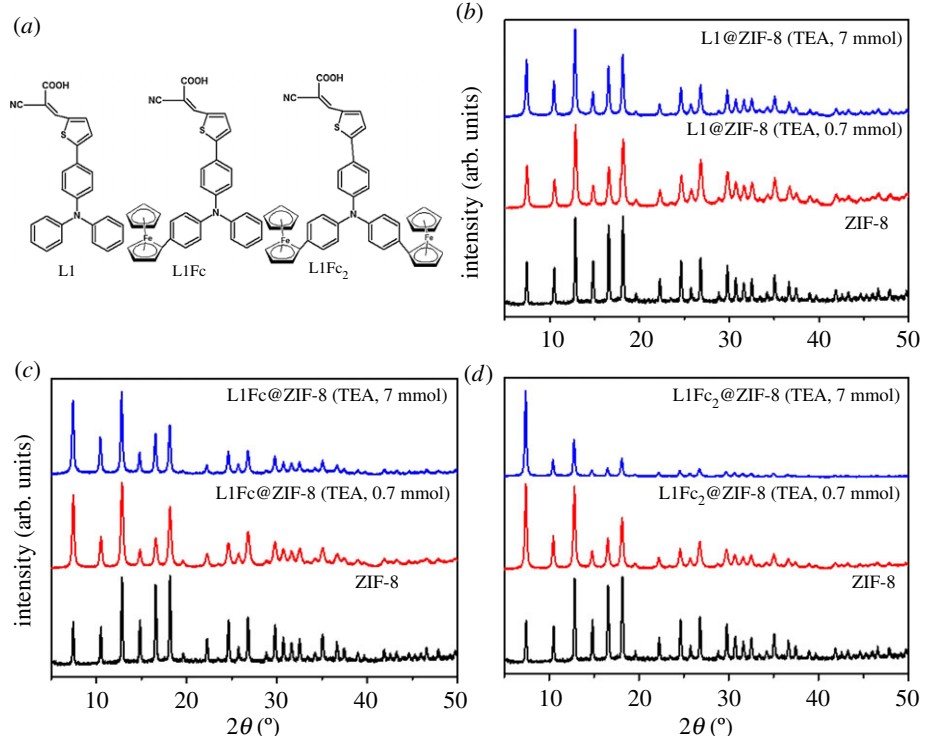

**Figure 1.** (a) Molecular structures of the encapsulated dyes, and (b–d) XRD patterns of the synthesized dye@ZIF-8 of L1 (b), L1Fc (c) and L1Fc$_2$ (d).

crystals as compared to that using only water [39,41]. XRD patterns confirm the formation of pure ZIF-8 phases (figure 1b–d) [42–44]. FT-IR spectra show peaks at 420 cm$^{-1}$ corresponding to Zn—N (electronic supplementary material, figure S1). The spectra show that the vibration and rotation modes (less than 1600 cm$^{-1}$) of the free dye are significantly suppressed due to the encapsulation. The difference between encapsulated and adsorbed dye can be illustrated using FT-IR spectra (electronic supplementary material, figure S2). Conventional microporous ZIF-8 has a small pore size (the largest cavity diameter and pore limiting diameter are 11.4 Å and 3.4 Å, respectively) compared to the dye, thus the dye cannot be located into the pore via adsorption. The band of O–H (3000 cm$^{-1}$) of carboxylic groups in the L1 dye is broad in the case of adsorbed dye compared to the encapsulated dye (electronic supplementary material, figure S2). The adsorbed dye displays a band at 1840 cm$^{-1}$ corresponding to cyano group C≡N, which is extremely weak in the encapsulated dye. There is also a shift in the wavenumber of the carbonyl group at 1580 cm$^{-1}$, and 1595 cm$^{-1}$ for encapsulated and adsorbed dye, respectively (electronic supplementary material, figure S2). These observations are due to association of the dye with the surface of ZIF-8. The band at 420 cm$^{-1}$ corresponding to Zn–N is strong in the case of encapsulation compared to L1 adsorbed (electronic supplementary material, figure S2). Based on sulfur atom percentage, as determined from elemental analysis, the encapsulated dye concentrations were determined to be 1.3%, 1.8% and 2.5% for L1@ZIF-8, L1Fc@ZIF-8 and L1Fc$_2$@ZIF-8, respectively.

TEM images of the prepared materials show that the crystal size of ZIF-8 ranges from 20 to 100 nm (figure 2), and decreases with the increase of dye size (L1Fc$_2$ > L1Fc > L1 in size due to ferrocene moieties, figure 1a). As previously reported, the addition of TEA to a solution of Zn(NO$_3$)$_2$ results in the formation of ZnO nanocrystals [39]. Addition of the dye to the formed ZnO leads to adsorption prior to conversion to ZIF-8 crystals after addition of Hmim. The large dyes such as L1Fc$_2$ may improve the dispersion of the formed ZnO nanocrystals and increase the nucleation rate over the growth rate of ZIF-8 crystal, resulting in small particles of ZIF-8. In addition, inter-particle mesopore structures are formed in L1Fc$_2$@ZIF-8 (figure 2). SEM images show that the particle size of dye@ZIF-8 for L1, L1Fc and L1Fc$_2$ ranges from 20 to 100 nm (electronic supplementary material, figure S3), which agrees with TEM images (figure 2). The colour of the synthesized materials confirms the presence of the dye in the final products, as shown in electronic supplementary material, figure S4.

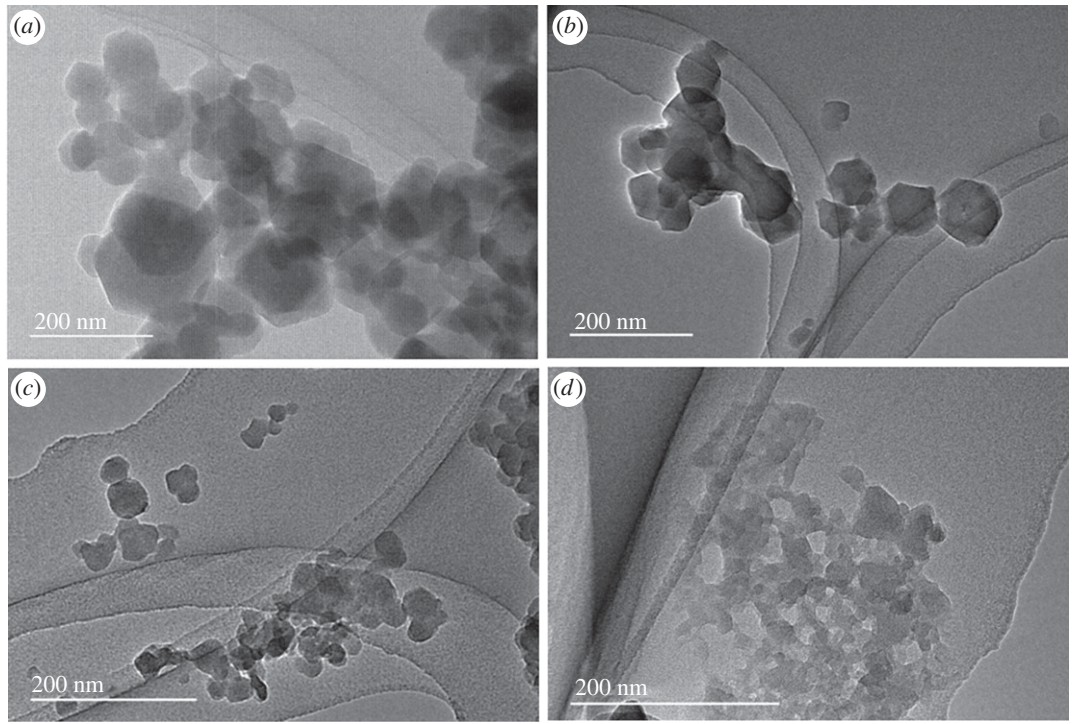

**Figure 2.** TEM image of (*a*) ZIF-8, (*b*) L1@ZIF-8, (*c*) L1Fc@ZIF-8 and (*d*) L1Fc$_2$@ZIF-8.

Results from EDX analyses reveal the presence of elements C, N, O, Zn and more importantly S from the dyes in ZIF-8 (figure 3*f*). Although a strong signal from Fe is also observed, it cannot be used as evidence for the presence of the dye because this peak could appear due to stray X-rays in TEM instruments. By contrast, the peak from S confirms the presence of the dyes in the ZIF-8 agglomerate. We thus consider it evidence in favour of the presence of the dye in the same region as the ZIF-8 crystals (figure 3). The spatial distribution of C, N, O and Zn was figured out using EELS spectrum imaging, and maps from this investigation are presented in figure 3*b−e*. These maps were generated by first applying weighted principal component analysis to the data cube and selecting the five components of highest variance for reconstruction. The maps from the signals over an energy window of 30 eV are presented in figure 3*b−e*. Of note is the presence of what appears to be a series of pores in the O, Zn and C maps, which we interpret to be the ZIF-8 mesoporous network. The distribution of O and Zn is strongly correlated within the MOF structure (correlation coefficient +0.88). Oxygen is not expected to be present in the crystal structure of ZIF-8 or its pore under high vacuum, while the dye has oxygen, which we interpret to mean that the dye has been incorporated directly into the ZIF-8 framework.

The porosity of the prepared material was estimated using N$_2$ adsorption−desorption isotherms (figure 4*a,b*). The surface areas extracted using the methods of Brunauer−Emmett−Teller and Langmuir are tabulated in electronic supplementary material, table S1 [45]. The surface area, pore volume and pore size increase with the increase of dye size (L1Fc$_2$ > L1Fc > L1, figure 4; electronic supplementary material, table S1), as determined using the Barrett−Joyner−Halenda method (BJH, figure 4*b*) and non-local density functional theory (electronic supplementary material, figure S5). The results reveal the presence of mesopores, i.e. hierarchical porous ZIF-8, and the pore size ranges from 5 to 20 nm with the majority at approximately 15 nm. The creation of mesopore is due to the dye template molecules.

## 3.2. Emission lifetime for free and encapsulated dyes

The free and encapsulated dyes show similar profiles for emission and excitation spectra (electronic supplementary material, figure S6). However, the excitation and emission signals for solid free dyes L1, L1Fc and L1Fc$_2$ are much noisier than those for the encapsulated ones due to aggregation and ferrocene moieties [6].

The emission lifetime measurements for the encapsulated dyes (L1, L1Fc and L1Fc$_2$, figure 5) into ZIF-8 were performed using TCSPC. All the emission intensities for the solid powders versus time

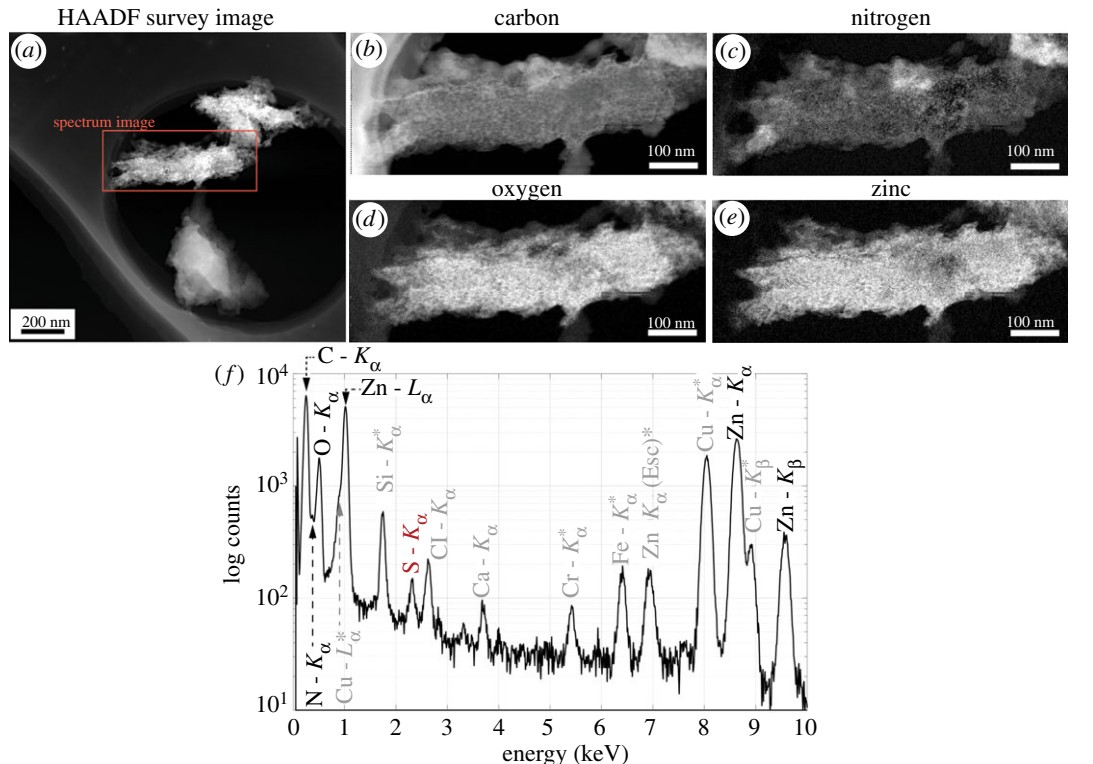

**Figure 3.** (a) HAADF image and (b−e) elemental spatial distribution of carbon (b), nitrogen(c), oxygen (d) and zinc (e), acquired using EELS spectrum imaging. (f) EDX spectrum simultaneously acquired from the same region. The peaks caused by the TEM system and surface contamination from the sample preparation process (i.e. Ca and Cl peaks) are marked in grey.

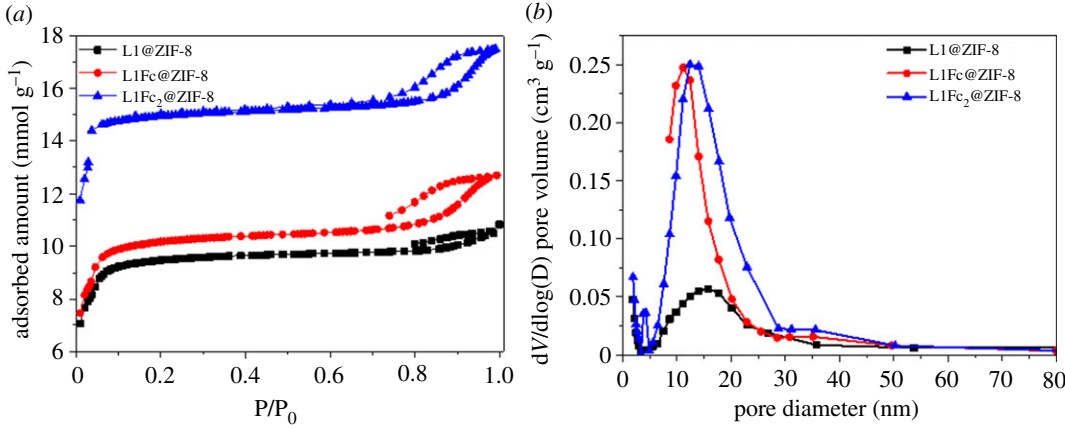

**Figure 4.** (a) $N_2$ adsorption−desorption of dye@ZIF-8 and (b) pore size distribution using BJH method.

showed multi-exponential components, due to the heterogeneity present in the solid phase. Thus, the average lifetimes were calculated to facilitate the comparison. All the average lifetimes for the solid forms of the dyes are shorter as compared to the corresponding dyes in solution, due to aggregation and twisting motion present in such dyes [6]. However, upon encapsulation of these dyes in ZIF-8, the corresponding emission lifetimes increase as follows: L1 (approx. 0.35 ns), L1@ZIF-8 (approx. 3.1 ns), L1Fc@ZIF-8 (approx. 2.6 ns) and L1Fc$_2$@ZIF-8 (1.75 ns). The lifetimes for solid powders of L1Fc and L1Fc$_2$ are shorter than the laser pulse width (IRF ≈ 70 ps) and could not be measured. The shorter lifetimes for encapsulated L1Fc and L1Fc$_2$ dyes are attributed to the charge transfer character present from Fc moiety to the L1 unit [6]. The base concentrations (0.7–7 mmol) have a negligible effect on the emission lifetime values for the encapsulated dyes. An increase of lifetime (greater than 4–8-fold) is observed for L1 and L1@ZIF-8, which is attributed to the suppression of both solid aggregations and twisting motions of the encapsulated dye.

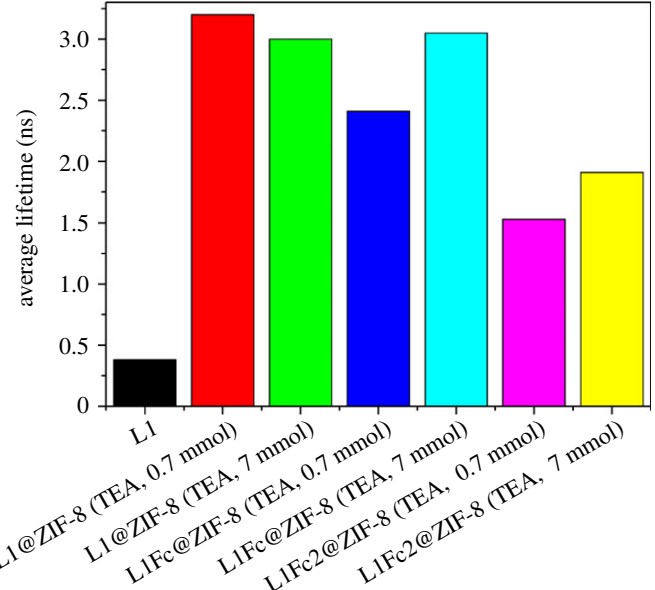

**Figure 5.** Excited state lifetime measurements of dyes encapsulated into ZIF-8.

**Table 1.** Summary of DSSC measurements for L1@ZIF-8.

| synthesis method | $V_{OC}$ (mV) | $J_{SC}$ (mA cm$^{-2}$) | FF (%) | Eff. (%) |
|---|---|---|---|---|
| *in-situ* growth (1 h) | 565 | 0.31 | 68.8 | 0.12 |
| *in-situ* growth (2 h) | 580 | 0.27 | 69.1 | 0.11 |
| *in-situ* growth (3 h) | 575 | 0.40 | 54.7 | 0.13 |
| direct disposition | 675 | 0.63 | 62.1 | 0.27 |
| layer by layer | 680 | 2.57 | 75.2 | 1.31 |

## 3.3. Solar cells based on the dye@ZIF-8 materials

The dye encapsulated ZIF-8 coated onto conductive glass substrates (electronic supplementary material, figure S7) were characterized using FT-IR (electronic supplementary material, figure S8), and SEM (electronic supplementary material, figure S9). FT-IR spectra confirm the presence of L1@ZIF-8 on the glass substrates (electronic supplementary material, figure S7). SEM images show a layer of the L1@ZIF-8 on top of a supporting TiO$_2$ layer with a thickness of 1–5 μm (electronic supplementary material, figure S9).

The DSSC parameters from devices based on L1@ZIF-8 materials in terms of $V_{OC}$ (mV), short circuit current density (Jsc, mA · cm$^{-2}$), fill factor (*FF*) and efficiencies ($\eta$) are tabulated in table 1. The typical $V_{OC}$ of standard TiO$_2$-based DSSCs for I$^-$-based system are known to be between 700 and 800 mV [30]. However, the working electrodes loaded with L1@ZIF-8 show a lower $V_{OC}$ (table 1). The highest $V_{OC}$ value was observed for cells based on a dye@ZIF-8 deposited using a layer-by-layer method (see experimental section), followed by a direct deposition method. The $V_{OC}$ value increases with the increase of the reaction time. This observation is in agreement with the previous study in [30]. Results show that the efficiency depends on the deposition method: layer-by-layer > direct deposition > *in-situ* growth. This can be linked to the higher loading of L1@ZIF-8 obtained by using the layer-by-layer method as compared to the other methods (electronic supplementary material, figure S7). The incident photon-to-electron conversion efficiency (IPCE) spectra for L1Fc, and L1Fc@ZIF-8 are shown in electronic supplementary material, figure S10. The IPCE spectrum of L1Fc shows a strong photoelectric response to ultraviolet light in the range from 400 to 600 nm, with the highest IPCE at 465 nm. A shift toward a shorter wavelength was observed for L1Fc@ZIF-8. The change of the shape of the IPCE spectrum for L1Fc@ZIF-8 may point to morphological alterations in the light absorbing dye due to encapsulation. It is not clear if the low performance of the systems in this study is caused by the low

dye loading or due to the absence of direct electron injection mechanisms from the dye to the metal oxide substrate [46,47]. The Nyquist plot of L1Fc@ZIF-8 shows a very high resistance for the encapsulated dye as compared to the free dye L1Fc (287 Ω) [6], due to the low conductivity of ZIF-8 (electronic supplementary material, figure S11). Encapsulation of the dye molecules limits their ability to uniformly bind to the semiconductor surface, affecting the electron transfer process at the photoanode. Further planned studies may explain the low efficiency and could improve the material performance.

# 4. Conclusion

Triethylamine-assisted synthesis of hierarchical porous ZIF-8 and a one-pot encapsulation of solar cell dyes were successfully achieved. The pore structure of the synthesized materials is tuneable and depends on the size of the encapsulated dye molecules. The encapsulated dyes show longer emission lifetimes compared to the corresponding non-encapsulated dyes, however, they tend to have lower DSSCs efficiencies. The encapsulation of solar cell dyes into MOFs may open further exploration for high DSSCs efficiencies.

Data accessibility. This article does not contain any additional data. All data are present in figures 1–5 in the main text and electronic supplementary material figures S1–S11.

Authors' contributions. H.N.A. developed and designed the project. He synthesized the materials and characterized them using TEM, SEM, XRD, FT-IR and $N_2$ adsorption–desorption isotherms. He performed data analysis and wrote the paper. J.C., M.K., L.K. synthesis the dyes and reported the DSSC performance. T.T. reported HAADF data. A.E. reported the lifetime measurements. H.N.A., X.Z. and J.C. revised the text. All authors discussed the results, commented on the manuscript and approve it.

Competing interests. Authors declare that there is no conflict of interests.

Funding. This work was funded by the Swedish Research Council (VR-NT). We thank the Knut and Alice Wallengberg Foundation for a grant for purchasing the electron microscopes.

Acknowledgements. We thank Dr Moataz Dowaidar (SU), Mr Sahar Sultan (SU) and Dr Ahmed F. Abdel-Magied (KTH) for the help.

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
