## [Reviewer comments · Royal Society Open Science]

Review History

RSOS-181370.R0 (Original submission)

Review form: Reviewer 1

Is the manuscript scientifically sound in its present form?

Yes

Are the interpretations and conclusions justified by the results?

No

Is the language acceptable?

Yes

Is it clear how to access all supporting data?

Yes

Do you have any ethical concerns with this paper?

No

Have you any concerns about statistical analyses in this paper?

No

Recommendation?

Major revision is needed (please make suggestions in comments)

Comments to the Author(s)

It is an interesting work. However, the qualities of MS should be improved before being published.

- In paper, there is little explanations of reason. For example, from TEM images, the particle size of dye@ZIF-8 decreased after adding dye, why?; suggesting adding size distribution.
 - In XRD patterns, the peak intensity of L1FC2@ZIF-8 is lower compared to others at 20-50o, why?
- S
- Authors state that, "the color of the synthesized materials confirms the presence of the dye in the final products", whether is there any other evidences to determine dye in the final products?
 - Emission lifetimes and DSSCs efficiencies in MS should be compared with the reported references.

. Fig.3f is unclear and its quality must be improved.

Review form: Reviewer 2

Is the manuscript scientifically sound in its present form?

No

Are the interpretations and conclusions justified by the results?

No

Is the language acceptable?

Yes

Is it clear how to access all supporting data?

Yes

Do you have any ethical concerns with this paper?

No

Have you any concerns about statistical analyses in this paper?

No

Recommendation?

Reject

Comments to the Author(s)

This manuscript reported a methodology of dye encapsulation for dye sensitized solar cells, where ZIF-8 was employed as dye host. This method enhanced the longer emission lifetimes of dye but at a cost of efficiency. Considering the novelty, quality and solidity of this manuscript,

the manuscript can NOT be recommended for publication.

Suggestions and questions for improvement of this manuscript are listed as below:

1. Introduction should be focused on gap point of this manuscript, instead of narrating the cumbersome information, which would confuse the readers.
2. Afterall, the core novelty of this manuscript lies in the encapsulation of dye into ZIF-8. Solid evidence that verify the successful encapsulation of dye into ZIF-8 should be provided and explicitly highlighted. EDX analyses are definitely not enough to draw the conclusion of successful encapsulation of dye. Same results could easily be obtained from the physical mixture of dyes and ZIF-8.

Decision letter (RSOS-181370.R0)

25-Feb-2019

Dear Dr Abdelhamid:

Manuscript ID: RSOS-181370

Title: "Towards Implementing Hierarchical Porous Zeolitic Imidazolate Frameworks in Dye Sensitized Solar Cells"

Thank you for submitting the above manuscript to Royal Society Open Science. Your paper was sent to reviewers and their comments are included at the bottom of this letter.

In view of the concerns raised by the reviewers, the manuscript has been rejected in its current form. However, a new manuscript may be submitted which takes into consideration these comments.

Please note that resubmitting your manuscript does not guarantee eventual acceptance, and that your resubmission will be subject to peer review before a decision is made.

Your resubmitted manuscript should be submitted by 25-Aug-2019. If you are unable to submit by this date please contact the Editorial Office.

Royal Society of Chemistry
Thomas Graham House
Science Park, Milton Road

Cambridge, CB4 0WF
Royal Society Open Science - Chemistry Editorial Office

On behalf of the Subject Editor Professor Anthony Stace and the Associate Editor Professor Claire Carmalt

REVIEWER(S) REPORTS:

Associate Editor Comments to Author ():

RSC Associate Editor:

Comments to the Author:

(There are no comments.)

RSC Subject Editor:

Comments to the Author:

(There are no comments.)

Reviewers' Comments to Author:

Reviewer: 1

Comments to the Author(s)

It is an interesting work. However, the qualities of MS should be improved before being published.

- In paper, there is little explanations of reason. For example, from TEM images, the particle size of dye@ZIF-8 decreased after adding dye, why?; suggesting adding size distribution.

- In XRD patterns, the peak intensity of L1FC2@ZIF-8 is lower compared to others at 20-50o, why?
S

- Authors state that, "the color of the synthesized materials confirms the presence of the dye in the final products", whether is there any other evidences to determine dye in the final products?

- Emission lifetimes and DSSCs efficiencies in MS should be compared with the reported references.

. Fig.3f is unclear and its quality must be improved.

Reviewer: 2

Comments to the Author(s)

This manuscript reported a methodology of dye encapsulation for dye sensitized solar cells, where ZIF-8 was employed as dye host. This method enhanced the longer emission lifetimes of dye but at a cost of efficiency. Considering the novelty, quality and solidity of this manuscript, the manuscript can NOT be recommended for publication.

Suggestions and questions for improvement of this manuscript are listed as below:

1. Introduction should be focused on gap point of this manuscript, instead of narrating the cumbersome information, which would confuse the readers.

2. Afterall, the core novelty of this manuscript lies in the encapsulation of dye into ZIF-8. Solid evidence that verify the successful encapsulation of dye into ZIF-8 should be provided and explicitly highlighted. EDX analyses are definitely not enough to draw the conclusion of successful encapsulation of dye. Same results could easily be obtained from the physical mixture of dyes and ZIF-8.

Author's Response to Decision Letter for (RSOS-181370.R0)

See Appendix A.

RSOS-190723.R0

Review form: Reviewer 1

Is the manuscript scientifically sound in its present form?

Yes

Are the interpretations and conclusions justified by the results?

Yes

Is the language acceptable?

Yes

Is it clear how to access all supporting data?

Yes

Do you have any ethical concerns with this paper?

No

Have you any concerns about statistical analyses in this paper?

No

Recommendation?

Accept with minor revision (please list in comments)

Comments to the Author(s)

"TEM images of the prepared materials show that the crystal size of ZIF-8 ranges from 20 to 100 nm (Figure 2), and decreases with the increase of dye size (L1Fc2> L1Fc>L1 in size, Figure 1a)."
For these results, can you tell much more reasons of size change due to the difference in dye size?
Or you can cite one paper to support them

Review form: Reviewer 2

Is the manuscript scientifically sound in its present form?

Yes

Are the interpretations and conclusions justified by the results?

Yes

Is the language acceptable?

Yes

Is it clear how to access all supporting data?

Yes

Do you have any ethical concerns with this paper?

No

Have you any concerns about statistical analyses in this paper?

No

Recommendation?

Accept as is

Comments to the Author(s)

I am satisfied with the revisions that the authors have made. The manuscript can be accepted for publication as it is.

Decision letter (RSOS-190723.R0)

10-May-2019

Dear Dr Abdelhamid:

Title: Towards Implementing Hierarchical Porous Zeolitic Imidazolate Frameworks in Dye Sensitized Solar Cells

Manuscript ID: RSOS-190723

Thank you for submitting the above manuscript to Royal Society Open Science. On behalf of the Editors and the Royal Society of Chemistry, I am pleased to inform you that your manuscript will be accepted for publication in Royal Society Open Science subject to minor revision in accordance with the referee suggestions. Please find the reviewers' comments at the end of this email.

The reviewers and handling editors have recommended publication, but also suggest some minor revisions to your manuscript. Therefore, I invite you to respond to the comments and revise your manuscript.

Because the schedule for publication is very tight, it is a condition of publication that you submit the revised version of your manuscript before 19-May-2019. Please note that the revision deadline will expire at 00.00am on this date. If you do not think you will be able to meet this date please let me know immediately.

1) A text file of the manuscript (tex, txt, rtf, docx or doc), references, tables (including captions) and figure captions. Do not upload a PDF as your "Main Document".

- 2) A separate electronic file of each figure (EPS or print-quality PDF preferred (either format should be produced directly from original creation package), or original software format)
- 3) Included a 100 word media summary of your paper when requested at submission. Please ensure you have entered correct contact details (email, institution and telephone) in your user account
- 4) Included the raw data to support the claims made in your paper. You can either include your data as electronic supplementary material or upload to a repository and include the relevant doi within your manuscript
- 5) All supplementary materials accompanying an accepted article will be treated as in their final form. Note that the Royal Society will neither edit nor typeset supplementary material and it will be hosted as provided. Please ensure that the supplementary material includes the paper details where possible (authors, article title, journal name).

Best wishes,

Dr Laura Smith
Publishing Editor, Journals

On behalf of the Subject Editor Professor Anthony Stace and the Associate Editor Professor Claire Carmalt.

RSC Associate Editor
Comments to the Author:
(There are no comments.)

Reviewer comments to Author:
Reviewer: 2

Comments to the Author(s)
I am satisfied with the revisions that the authors have made. The manuscript can be accepted for publication as it is.

Reviewer: 1

Comments to the Author(s)

"TEM images of the prepared materials show that the crystal size of ZIF-8 ranges from 20 to 100 nm (Figure 2), and decreases with the increase of dye size (L1Fc2> L1Fc>L1 in size, Figure 1a)."

For these results, can you tell much more reasons of size change due to the difference in dye size? Or you can cite one paper to support them

Author's Response to Decision Letter for (RSOS-190723.R0)

See Appendix B.

RSOS-190723.R1 (Revision)

Review form: Reviewer 1

Is the manuscript scientifically sound in its present form?

Yes

Are the interpretations and conclusions justified by the results?

Yes

Is the language acceptable?

Yes

Is it clear how to access all supporting data?

Yes

Do you have any ethical concerns with this paper?

No

Have you any concerns about statistical analyses in this paper?

No

Recommendation?

Accept as is

Comments to the Author(s)

Recommend to be published in present form.

Decision letter (RSOS-190723.R1)

04-Jun-2019

Dear Dr Abdelhamid:

Title: Towards Implementing Hierarchical Porous Zeolitic Imidazolate Frameworks in Dye Sensitized Solar Cells

Manuscript ID: RSOS-190723.R1

It is a pleasure to accept your manuscript in its current form for publication in Royal Society Open Science. The chemistry content of Royal Society Open Science is published in collaboration with the Royal Society of Chemistry.

RSC Associate Editor:
Comments to the Author:
(There are no comments.)

RSC Subject Editor:
Comments to the Author:
(There are no comments.)

Reviewer(s)' Comments to Author:
Reviewer: 1

Comments to the Author(s)
Recommend to be published in present form.

Appendix A

Manuscript ID: RSOS-181370

Title: "Towards Implementing Hierarchical Porous Zeolitic Imidazolate Frameworks in Dye Sensitized Solar Cells"

Dear Editor,

Many thanks for your decision and email (15 Feb 2019) regarding our manuscript ID RSOS-181370. We addressed the comments from reviewer's as shown in blue color as below.

REVIEWER(S) REPORTS:

Associate Editor Comments to Author ():

RSC Associate Editor:

Comments to the Author:

(There are no comments.)

RSC Subject Editor:

Comments to the Author:

(There are no comments.)

Reviewers' Comments to Author:

Reviewer: 1

Comments to the Author(s)

It is an interesting work. However, the qualities of MS should be improved before being published.

Response: Many thanks for your support. The manuscript has been improved.

- In paper, there is little explanations of reason. For example, from TEM images, the particle size of dye@ZIF-8 decreased after adding dye, why?; suggesting adding size distribution.

Response: The decrease of particle size is mainly due to the increase of the dye size.

- In XRD patterns, the peak intensity of L1FC₂@ZIF-8 is lower compared to others at 20-50°, why? S

Response: This could be due to the X-ray absorption of iron that present in L1FC₂.

- Authors state that, “the color of the synthesized materials confirms the presence of the dye in the final products”, whether is there any other evidences to determine dye in the final products?

Response: Yes, the dye concentration was determined using elemental analysis. HAADF, EELS and EDX analysis was also used to confirm the presence of the dye.

- Emission lifetimes and DSSCs efficiencies in MS should be compared with the reported references.

Response: Thanks for suggestion. It was added.

Fig.3f is unclear and its quality must be improved.

Response: Improved accordingly.

Figure 3 a) HAADF image and b-e) elemental spatial distribution of carbon (b), nitrogen(c), oxygen (d), and zinc (e), acquired using EELS spectrum imaging. f) EDX spectrum simultaneously acquired from the same region. The peaks caused by the TEM system and surface contamination from the sample preparation process (i.e. Ca and Cl peaks) are marked in grey.

Reviewer: 2

Comments to the Author(s)

This manuscript reported a methodology of dye encapsulation for dye sensitized solar cells, where ZIF-8 was employed as dye host. This method enhanced the longer emission lifetimes of dye but at a cost of efficiency. Considering the novelty, quality and solidity of this manuscript, the manuscript can NOT be recommended for publication. Suggestions and questions for improvement of this manuscript are listed as below:

1. Introduction should be focused on gap point of this manuscript, instead of narrating the cumbersome information, which would confuse the readers.

Response: Thanks for suggestion. Introduction has been revised accordingly.

2. Afterall, the core novelty of this manuscript lies in the encapsulation of dye into ZIF-8. Solid evidence that verify the successful encapsulation of dye into ZIF-8 should be provided and explicitly highlighted. EDX analyses are definitely not enough to draw the conclusion of successful encapsulation of dye. Same results could easily be obtained from the physical mixture of dyes and ZIF-8.

Response: we have several evidences. The dye's size is larger than the samll pore size of conventional microporous ZIF-8 (Largest Cavity Diameter, and Pore Limiting Diameter are 11.4 Å, and 3.4 Å, respectively). First evidence, there is no observation of the dye in the surface of ZIF-8 in TEM images. Second, nitrogen adsorption reveals the formation of mesopore due to the dye. Third, we added a new data set of FT-IR showing the difference between encapsulated and adsorbed dye.

" The spectra show that the vibration and rotation modes ($<1600\text{ cm}^{-1}$) of the free dye are significantly suppressed due to the encapsulation. The difference between encapsulated and adsorbed dye can be illustrated using FT-IR spectra (Figure S2). Conventional microporous ZIF-8 has small pore size compared to the dye, thus the dye cannot be located into the pore via adsorption. The band of O—H (3000 cm^{-1}) of carboxylic groups in the L1 dye is broad in case of adsorbed dye compared to the encapsulated dye (Figure S2). The adsorbed dye display a band at 1840 cm^{-1} corresponding to cyano group $\text{C}\equiv\text{N}$, which is absent in the encapsulated dye. There is also shift in the wavenumber of carbonyl group at 1580 cm^{-1} , and 1595 cm^{-1} for encapsulated and adsorbed dye, respectively (Figure S2). These observations are due to association of the dye with the surface of ZIF-8. Based on sulphur atom percentage, as determined from elemental analysis, the encapsulated dye concentrations were determined to be 1.3%, 1.8%, and 2.5% for L1@ZIF-8, L1Fc@ZIF-8, and L1Fc₂@ZIF-8, respectively."

Figure S2 FT-IR spectra of L1 encapsulated and adsorbed into ZIF-8.

Appendix B

Manuscript ID: RSOS-181370

Title: "Towards Implementing Hierarchical Porous Zeolitic Imidazolate Frameworks in Dye Sensitized Solar Cells"

Dear Editor,

Many thanks for your decision and email (10-May-2019) regarding our manuscript ID RSOS-181370. We addressed the comments from reviewers as shown in blue color as below.

Response: Uploaded.

1) A text file of the manuscript (tex, txt, rtf, docx or doc), references, tables (including captions) and figure captions. Do not upload a PDF as your "Main Document".

Response: A word file containing the figures has been uploaded.

2) A separate electronic file of each figure (EPS or print-quality PDF preferred (either format should be produced directly from original creation package), or original software format)

Response: Uploaded.

3) Included a 100 word media summary of your paper when requested at submission. Please ensure you have entered correct contact details (email, institution and telephone) in your user account

Response: Followed.

4) Included the raw data to support the claims made in your paper. You can either include your data as electronic supplementary material or upload to a repository and include the relevant doi within your manuscript

Response: An electronic supplementary material is also uploaded.

5) All supplementary materials accompanying an accepted article will be treated as in their final form. Note that the Royal Society will neither edit nor typeset supplementary material and it will be hosted as provided. Please ensure that the supplementary material includes the paper details where possible (authors, article title, journal name).

Response: Followed

RSC Associate Editor

Comments to the Author:

(There are no comments.)

Reviewer comments to Author:

Reviewer: 2

Comments to the Author(s)

I am satisfied with the revisions that the authors have made. The manuscript can be accepted for publication as it is.

Response: Many thanks for your support.

Reviewer: 1

Comments to the Author(s)

"TEM images of the prepared materials show that the crystal size of ZIF-8 ranges from 20 to 100 nm (Figure 2), and decreases with the increase of dye size (L1Fc2>L1Fc>L1 in size, Figure 1a)." For these results, can you tell much more reasons of size change due to the difference in dye size? Or you can cite one paper to support them

Response: Many thanks for suggestion. References and explanation have been added as highlighted in red color.

"TEM images of the prepared materials show that the crystal size of ZIF-8 ranges from 20 to 100 nm (Figure 2), and decreases with the increase of dye size (L1Fc₂> L1Fc>L1 in size due to ferrocene moieties, Figure 1a). As previously reported, the addition of TEA to a solution of Zn(NO₃)₂ results in the formation of ZnO nanocrystals³⁹. Addition of the dye to the formed ZnO leads to adsorption prior to conversion to ZIF-8 crystals after addition of Hmim. The large dyes such as L1Fc₂ may improve the dispersion of the formed ZnO nanocrystals and increase the nucleation rate over than the growth rate of ZIF-8 crystal resulting to small particles of ZIF-8. In addition, inter-particle mesopore structures are formed in L1Fc₂@ZIF-8 (Figure 2). SEM images show that the particle size of dye@ZIF-8 for L1, L1Fc, and L1Fc₂ ranges from 20 to 100 nm (Figure S3), which agrees with TEM images (Figure 2). The color of the synthesized materials confirms the presence of the dye in the final products, as shown in Figure S4."